# Development and Content Validation of a Person-Centered Care Instrument for Healthcare Providers

**DOI:** 10.3390/nursrep15100355

**Published:** 2025-10-02

**Authors:** Krishan Soriano, Sora Nakatani, Kaito Onishi, Hirokazu Ito, Youko Nakano, Yoshiyuki Takashima, Yueren Zhao, Allan Paulo Blaquera, Ryuichi Tanioka, Feni Betriana, Gil Platon Soriano, Yuko Yasahura, Kyoko Osaka, Matsuko Kataoka, Misao Miyagawa, Masashi Akaike, Minoru Irahara, Tetsuya Tanioka

**Affiliations:** 1Graduate School of Health Sciences, Tokushima University, Tokushima 770-8509, Japan; ksoriano@spup.edu.ph (K.S.); kai10.onishi10@gmail.com (K.O.); 2Major in Nursing, School of Health Sciences, Faculty of Medicine, Tokushima University, Tokushima 770-8509, Japan; c202203023@tokushima-u.ac.jp (S.N.); misaomiyagawa2023@gmail.com (M.M.); 3Graduate School of Biomedical Sciences, Tokushima University, Tokushima 770-8509, Japan; h.itoh@tokushima-u.ac.jp (H.I.); yasuhara@tokushima-u.ac.jp (Y.Y.); akaike.masashi@tokushima-u.ac.jp (M.A.); irahara@tokushima-u.ac.jp (M.I.); 4Department of Nursing, Nursing Course of Kochi Medical School, Kochi University, Kochi 783-8505, Japan; nakano.yoko.rg@kochi-u.ac.jp (Y.N.); osaka@kochi-u.ac.jp (K.O.); 5Faculty of Nursing, School of Medicine, Nara Medical University, Kashihara 634-0813, Japan; y-takashima@naramed-u.ac.jp; 6Department of Psychiatry, Fujita Health University, Toyoake 470-1192, Japan; taipeeyen@gmail.com; 7School of Nursing and Allied Health Sciences, St. Paul University Philippines, Tuguegarao City 3500, Philippines; ablaquera@spup.edu.ph; 8Faculty of Health Sciences, Hiroshima Cosmopolitan University, Hiroshima 731-3166, Japan; tanioka@hcu.ac.jp; 9Center for Biomedical Research, National Research and Innovation Agency (BRIN), Cibinong 16911, Indonesia; feni004@brin.go.id; 10Department of Nursing, College of Allied Health, National University Philippines, Manila 1008, Philippines; gil.p.soriano@gmail.com; 11Mifune Hospital, Marugame 763-0073, Japan; kataoka@mifune-hp.jp

**Keywords:** person-centered care, instrument development, healthcare practice, patient-centered, quality improvement, psychometric validation, 9-point Likert scale agreement score

## Abstract

**Background/Objectives**: Despite the increasing recognition of person-centered care (PCC), existing evaluation tools often have profession-specific limitations, lacking broad applicability across interdisciplinary contexts. This study aimed to develop and validate the Person-Centered Care Instrument (PCCI), designed to assess the competence of healthcare providers from diverse professions. **Methods**: Using a two-round modified Delphi technique, ten experts validated an initial pool of 63 items. The process assessed both face validity (overall appropriateness) and content validity using a 9-point Likert scale and the Item-level Content Validity Index (I-CVI). Items with a median rating of 6 or higher and an I-CVI of ≥0.70 were retained. **Results**: The final PCCI consists of 37 items, with a scale-level content validity index of 0.65. Three items achieved universal agreement among the experts (I-CVI = 1.0). For the final 37-item PCCI, the Scale-level Content Validity Index (S-CVI) was 0.65, and the index based on universal agreement was 0.22. **Conclusions**: The developed PCCI demonstrated good face and content validity, making it a valid and broadly applicable tool for assessing competence in delivering PCC. This instrument can support quality improvement initiatives and help promote a culture of empathy and respect in healthcare.

## 1. Introduction

### 1.1. Theoretical Background

The limitations of the traditional biomedical model—which often focuses solely on illness or disability rather than the individual—have highlighted the critical need for a new approach in caring for vulnerable populations [1,2,3]. Person-centered care (PCC) has emerged as a foundational standard in response to the growing emphasis on quality and accountability within the healthcare field. This overarching concept, encompassing aspects such as “individualized-centered care” and “client-centered care,” emphasizes a holistic approach in which individuals are regarded as whole persons with complex needs, values, and goals, which are integrated into the care process [4]. This shift from the traditional biomedical model reflects a recognition that patient engagement plays a critical role in optimizing health outcomes.

The necessity of this shift is rooted in several critical areas.

First, PCC addresses the need to empower patients in their own health journeys. The approach places patients at the heart of healthcare decisions. It is based on the practices of treating everyone with respect and empathy, listening carefully, and sharing information [5]. Unlike traditional methods that focus mainly on symptoms and diagnoses, PCC considers the emotional, social, and psychological needs of a person as a whole. When patients and their families are engaged, they tend to adhere to their treatment plans and experience less stress. A previous study [6] shows that when patients participate in decisions, their satisfaction and quality of life improve. Thus, this approach also fosters trust between patients and healthcare providers.

Second, PCC is essential for building a therapeutic relationship based on empathy and compassion. Recognizing a patient’s feelings and understanding their situation and lived experiences leads to compassionate care. Showing compassion creates a safe space in which patients feel comfortable opening up and sharing their concerns [7]. The psychological safety fostered by this approach encourages honest conversations, resulting in improved healthcare quality and the alignment of care with patients’ true values, thereby enhancing satisfaction and quality of life.

As healthcare systems increasingly validate and adopt this approach, it becomes imperative to clearly define the parameters of PCC and establish objective measures for its implementation [8]. The growing recognition of the importance of PCC stems from its capacity to enhance healthcare safety, quality, and delivery, ultimately improving patients’ overall quality of life [9].

Finally, as healthcare systems increasingly face complex, chronic conditions and the need for long-term care, a one-size-fits-all approach is no longer sustainable. The PCC approach enables a healthcare practice that upholds the dignity and autonomy of care recipients, honors personal choices, and delivers tailored care that recognizes individuals as unique and independent [10]. By fostering healthy relationships, an effective care environment, and shared decision-making, the PCC approach leads to greater patient satisfaction and empowerment, improved treatment adherence, and enhanced trust between patients and providers [11,12]. Through the shared decision-making process, healthcare providers engage patients and their families. This empowers patients by giving them the autonomy to choose treatments that align with their lifestyles and values. Further, it fosters mutual trust and improves healthcare outcomes [13].

However, despite its recognized benefits, the effective implementation of PCC is often hindered by a lack of standardized parameters and accurate evaluation tools.

### 1.2. Existing PCC Scales

In response to the need for robust PCC evaluation, numerous instruments have been developed. Examples include the Person-Centered Care Assessment Tool (P-CAT) [14], which assesses staff perceptions of person-centeredness; the Person-Centered Climate Questionnaire (PCQ) [14,15], with patient (PCQ-P) and staff (PCQ-S) versions, which evaluates the care environment; and the Individualized Care Scale (ICS) [16], which captures care from both patient and nurse perspectives. Other notable instruments include the Person-Centered Practice Inventory–Staff (PCPI-S) [17], which assesses staff values based on McCormack and McCance’s framework [18], and the Consultation and Relational Empathy (CARE) measure, which focuses on empathy in clinical consultations.

Despite the array of existing tools, most are limited in scope, specifically targeting particular professional groups or narrow aspects of care [11,19]. This profession-specific focus presents a significant limitation in the contemporary healthcare landscape, in which interdisciplinary collaboration is essential for delivering holistic and effective care. Furthermore, few existing tools are comprehensive enough to integrate organizational, cultural, relational, and emotional aspects of care [4,20]. Their use in contexts that call for a more comprehensive understanding of PCC as a relational and systemic activity is impeded by this fragmented perspective. Many have also been criticized for lacking a comprehensive theoretical framework [18,21,22], which limits their generalizability and utility for assessing competence across diverse healthcare professions.

These shortcomings highlight a significant gap in the literature for a comprehensive, theory-based, multidisciplinary tool. By developing the Person-Centered Care Instrument (PCCI) to be both transdisciplinary and theoretically grounded, our study directly fills this gap, enabling its use in a variety of healthcare settings.

### 1.3. Conceptual Framework Development

Addressing the notable lack of existing measurement tools, this study is underpinned by a novel, conceptual framework for PCC designed to be broadly applicable to all types of healthcare providers. This comprehensive framework offers a robust foundation for assessing PCC practice across the full spectrum of healthcare professionals who collectively influence patient outcomes.

This framework was systematically developed, drawing upon established PCC models and dimensions (e.g., Picker Institute’s dimensions, McCormack and McCance’s Person-Centered Practice Framework, and other comprehensive reviews of PCC concepts) to synthesize core elements essential for a universal understanding of person-centered practice across healthcare disciplines [21,23,24]. This study identifies eight core concepts for a personalized and dignified care framework that emphasizes a holistic approach, structured around the following concepts.

#### 1.3.1. Concept 1: Respect and Empathy

This fundamental concept involves acknowledging patients’ dignity, validating their emotional experiences, and treating them as unique individuals [25,26,27]. It is foundational for establishing trust and open communication, emphasizing compassionate engagement.

#### 1.3.2. Concept 2: Partnership and Trust

This concept underscores the importance of collaborative relationships among healthcare providers, patients, and families, ensuring equal participation in care decisions and shared responsibility for health outcomes [19,28].

#### 1.3.3. Concept 3: Individualization and Consideration for Diversity

This concept highlights care tailored to account for biological, psychological, social, cultural, and environmental influences, recognizing that each patient’s background profoundly shapes their health needs and preferences [20,29].

#### 1.3.4. Concept 4: Shared Decision-Making

This concept focuses on promoting informed choices by supporting patients to actively participate in their care, clarify their goals, and make decisions that align with their values and circumstances [30].

#### 1.3.5. Concept 5: Emotional and Psychological Support

This concept focuses on reinforcing mental well-being by fostering security, confidence, and empowerment, essential for recovery and independence [24]. It also addresses patient anxiety, fear, and stress.

#### 1.3.6. Concept 6: Comprehensive Care and Holistic Perspective

This concept highlights the importance of treatment extending beyond physical ailments to include psychological, social, and spiritual dimensions, aligning with a biopsychosocial model of health and well-being [31,32,33].

#### 1.3.7. Concept 7: Effective Information Sharing with Care Recipients

This concept promotes health literacy by providing clear, accessible, and timely information, empowering patients to make informed and independent decisions about their care [34].

#### 1.3.8. Concept 8: Flexible Care

This concept highlights the importance of adaptive interventions, where healthcare professionals continually assess and adjust support based on patients’ evolving needs, responses, and changing circumstances [26]. It moves beyond rigid, protocol-driven approaches [35], underscoring the ability of care systems and individual care providers to innovate and adapt quickly to fluctuating patient needs and unpredictable conditions, ensuring timely and appropriate care delivery [36].

### 1.4. Research Objectives

Currently, a comprehensive and validated instrument capable of measuring PCC across the full spectrum of healthcare providers—including physicians, nurses, allied health professionals, and support staff—is lacking. The development of such a reliable instrument is essential for translating PCC principles into tangible improvements in healthcare delivery and patient outcomes [37].

Thus, the goal of this study is to develop and validate the PCCI for healthcare professionals and establish its content validity. By providing a thorough, theoretically based instrument that can assess PCC in a variety of transdisciplinary contexts, this work adds to the body of knowledge by directly addressing the gaps left by previous scales. It is anticipated that this tool will help direct educational initiatives, promote quality improvement, and strengthen a collaborative and empathetic culture in healthcare practice.

## 2. Materials and Methods

This study employed a two-phase process for the development and validation of the PCCI for healthcare providers, utilizing a modified Delphi technique and adhering to DELPHISTAR [38]—an established reporting standard for Delphi studies in social and health sciences—to ensure methodological rigor [39].

### 2.1. Study Design and Phases

The study comprised two phases: (1) instrument development and (2) face and content validation through a two-round modified Delphi study.

#### 2.1.1. Phase 1: Instrument Development

An initial pool of 63 items was created through a comprehensive review of existing PCC measures and related literature. The item pool developed in this study was designed to align with the eight conceptual domains of the research framework. The sources referenced for the items within each concept are as follows: (1) respect and empathy [25,26,27], (2) partnership and trust [19,28], (3) individualization and diversity [20,29], (4) shared decision-making [30], (5) emotional and psychological support [24], (6) comprehensive care [31,32,33], (7) effective information sharing [34], and (8) flexible care [26,35,36].

#### 2.1.2. Phase 2: Face and Content Validation

A two-round modified Delphi study was conducted to obtain expert consensus on the PCCI items. In Round 1, experts evaluated the overall appropriateness and relevance (face validity) of the initial 63 items, in addition to providing qualitative feedback and suggesting new items. Following revisions based on round 1 feedback, round 2 focused on establishing the content validity of the refined item set [40].

### 2.2. Statistical Analysis

Quantitative data from the expert ratings were analyzed using descriptive statistics, including median, minimum, and maximum values, to determine the level of consensus among the experts [41]. All data entry and calculations for the content validity indices were performed using Microsoft Excel^®^.

We chose to use a 9-point Likert scale in the modified Delphi method because it is a validated measure of consensus [42] that offers greater precision and nuance than shorter scales. This granularity allowed the experts to differentiate between similar levels of importance or agreement, leading to a more robust and detailed consensus [43,44].

Panelists rated each item’s relevance and importance using the Likert scale over two rounds of the Delphi process, with scores ranging from 1 to 9 (1 = Not at all important, 2 = Very unimportant, 3 = Not important, 4 = Somewhat unimportant, 5 = Neither important nor unimportant, 6 = Somewhat important, 7 = Important, 8 = Very important, and 9 = Extremely important). Prior to the calculation in which content validity index, rating was assigned a value of 1 to importance scores (6 to 9), and a value of 0 to scores from 1 to 5 [45].

The Item-level Content Validity Index (I-CVI) was calculated as the proportion of experts who rated an item as highly relevant (typically 6–9 on a 9-point scale). Universal agreement (UA) was assessed as the proportion of experts in perfect agreement on the highest relevance rating (e.g., all experts rating “9”). The scale-level content validity index (S-CVI), average method (S-CVI/Ave), was used to derive the mean of the I-CVI scores for all retained items. Similarly, the scale-level content validity index, universal agreement method (S-CVI/UA), was employed to evaluate the proportion of items for which all experts were in perfect agreement on the highest relevance rating.

Items were retained if they achieved a median relevance rating of 6 or higher and an I-CVI of ≥0.70. While Polit and Beck [46,47] recommend a more stringent I-CVI of 0.78 for three or more experts, we chose a threshold of 0.70 to allow for a larger item pool. This was a strategic decision to retain a greater number of items for subsequent exploratory and confirmatory factor analyses in future studies. Items that met this threshold were included in the PCCI for the planned large-scale survey.

## 3. Results

### 3.1. Expert Panel

The eligibility criteria were as follows: (1) holding a doctoral degree in medicine, health, or nursing; (2) having publications related to person-centered care; (3) having professional experience in improving healthcare services; and (4) having at least ten years of clinical experience with a master’s degree or higher.

The Delphi panel comprised 10 experts with extensive knowledge and experience in PCC. The panel included healthcare professionals specializing in nursing (n = 7), physiotherapy (n = 2), and medicine (n = 1), all holding a master’s degree or higher. Their clinical and educational experience averaged 27.5 years, with ages ranging from 30 to 60 years, ensuring a diverse and experienced perspective. Table 1 presents the characteristics of the expert panel.

### 3.2. Instrument Refinement and Content Validity

The Delphi process involved a panel of 10 experts who participated in two evaluation rounds.

#### 3.2.1. Round 1 Results

All 10 experts (100% response rate) completed Round 1.

The item pool was reduced from 63 to 60 items based on expert feedback regarding appropriateness and relevance (face validity) and initial quantitative ratings. Items were primarily excluded owing to redundancy or perceived lack of relevance.

Then, the 60 items were grouped under the eight previously established conceptual domains. The initial distribution of items per concept after Round 1 was as follows: “Respect and Empathy” (10 items), “Partnership and Trust” (5 items), “Individualization and Consideration of Diversity” (10 items), “Shared Decision-Making” (8 items), “Emotional and Psychological Support” (5 items), “Comprehensive Care and Holistic Perspective” (8 items), “Effective Information Sharing with Care Recipients” (8 items), and “Flexible Care” (6 items).

#### 3.2.2. Round 2 Results

All 10 experts (100% response rate) completed Round 2. Figure 1 shows the item development process. Experts rated the relevance of the 60 items using a 9-point Likert scale. Applying the retention criteria, a total of 37 items were retained for the final PCCI. Of the 60 items from Round 1, 23 items were deleted owing to low relevance ratings or lack of consensus.

Table 2 presents the final distribution of items across the eight conceptual domains.

## 4. Discussion

### 4.1. Content Validity

While the rigorous Delphi process yielded a robust set of 37 items, the lower-than-ideal S-CVI/Ave score of 0.65, and particularly the low S-CVI/UA score of 0.22 warrant discussion.

The S-CVI/Ave score, while below the commonly recommended threshold of 0.80 [46,47], reflects a solid overall relevance of the retained items. This score is likely influenced by the PCC construct’s inherent complexity, wherein some dimensions may have slightly lower average ratings owing to differing expert perspectives.

The low S-CVI/UA score is particularly noteworthy. This strict measure of universal agreement highlights that only a few items (3 out of 37) were considered “extremely important” by all experts. This is not necessarily a weakness of the instrument itself but a direct result of our panel’s interdisciplinary composition. Experts from various professional backgrounds (e.g., nursing, medicine, and physical therapy) may hold differing views on the precise importance of each item based on their specific clinical context [48,49]. This finding supports the need for a scale that is broadly applicable across professions, as it demonstrates that no single dimension of PCC is universally “most important” for all experts.

These results underscore that while the PCCI’s content is valid and relevant, further psychometric testing is essential. The mixed S-CVI values indicate the need for subsequent statistical analyses, such as exploratory and confirmatory factor analysis, to refine the scale’s structure and ensure it accurately captures the full scope of person-centered practice across diverse healthcare professions.

### 4.2. Comparison of Existing PCC Scale and Developed PCCI

While numerous validated instruments have been developed to measure PCC, most existing scales are limited to either specific professional groups or narrow dimensions of PCC. The PCCI contributes to the literature by offering a comprehensive, cross-disciplinary tool that captures eight distinct domains, thereby addressing gaps in earlier measures.

The PCCI’s domain of Respect and Empathy resonates with findings from the Consultation and Relational Empathy (CARE) Measure [50], which demonstrated a strong association between higher empathy scores and patient trust and satisfaction. However, the PCCI extends this concept by including competencies related to non-verbal communication and a contextual understanding of patients’ lived experiences, aspects less emphasized in prior tools.

The PCCI’s emphasis on Partnership and Trust, particularly through the collaborative involvement of patients and families, aligns with the Person-Centered Climate Questionnaire (PCQ-P and PCQ-S) [14,15]. Whereas the PCQ highlighted the role of the organizational climate, the PCCI reframes partnership as a measurable provider-level competency, offering actionable insights for training and professional development.

Furthermore, the PCCI’s dimension of Individualization and Consideration for Diversity is conceptually similar to the Individualized Care Scale (ICS) [16], which underscores the importance of tailoring care to a patient’s personal values and cultural background. While the ICS focused on nurse-patient dyads, the PCCI is designed for interdisciplinary applicability, enhancing its generalizability across healthcare contexts.

The PCCI’s focus on Shared Decision-Making parallels findings from both the PCQ [14,15] and studies on decision support [13], which demonstrate that collaborative involvement in care planning improves patient satisfaction and engagement. The PCCI competencies advance this by focusing on empowering patient autonomy and simplifying complex medical information.

The domain of Comprehensive Care and Holistic Perspective reflects themes captured in the Person-Centered Practice Inventory-Staff (PCPI-S) [17], which emphasized the integration of biopsychosocial care. The PCCI complements this by situating holistic care within the direct practice of providers, enabling a dual focus on organizational systems and individual competencies.

Finally, while effective communication is an established component of PCC, the PCCI’s domain of Effective Information Sharing [34] operationalizes this into concrete skills that can be systematically assessed across healthcare professions. Similarly, the Flexible Care dimension reflects McCormack and McCance’s (2006) framework [18] for responsive practice, but the PCCI uniquely translates this principle into provider-level adaptability, thereby bridging theory and practice.

Taken together, the PCCI not only converges with existing PCC scales in affirming the importance of empathy, individualization, shared decision-making, and holistic care but also introduces a novel, cross-disciplinary structure that captures provider competencies across the full spectrum of healthcare roles. This positions the PCCI as both complementary to and distinct from earlier instruments, providing a comprehensive measure that can support education, research, and quality improvement initiatives in diverse care contexts.

### 4.3. Interpretation of the PCCI Used in Practice

The PCCI is designed to serve as a vital tool for assessing the perceptions of healthcare providers regarding PCC. Interpreting scores on the PCCI, both overall and for individual concepts, offers critical insights into current attitudes and practices, providing a baseline for targeted improvement initiatives.

As for the evaluation method, the 6-point Likert scale (1 = Strongly disagree, 2 = Somewhat disagree, 3 = Slightly disagree, 4 = Slightly agree, 5 = Somewhat agree, 6 = Strongly agree) allows for a granular assessment.

Interpreting PCCI scores provides crucial insights for targeted improvement. High scores indicate a strong foundation in person-centered attitudes, suggesting these providers are ideal candidates for leadership roles. Conversely, mid-range scores suggest an opportunity for growth and a need for targeted education, while low scores highlight significant gaps requiring fundamental, intensive training.

#### 4.3.1. Practical and Clinical Significance of the PCCI

The practical and clinical significance of the PCCI extends beyond its content validation; it has the capacity to transform patient safety initiatives, quality improvement programs, and healthcare education.

First, by emphasizing critical skills, such as empathy, respect, and shared decision-making, the PCCI provides a systematic framework for guiding curriculum development in healthcare education. Educators can use PCCI results to create targeted training modules that foster relational and communication skills in undergraduate and graduate programs, ensuring alignment with PCC principles from early professional development [21,22].

Second, the PCCI offers a valuable resource for professional development and in-service training. Organizations can assess the strengths and weaknesses of clinical teams by evaluating staff attitudes and behaviors across its eight dimensions. These insights can help managers design targeted workshops and ongoing training to address identified gaps. Prior research indicates that structured interventions based on PCC frameworks can enhance patient outcomes, collaboration, and employee engagement [11,19].

Third, the PCCI has the potential to enhance patient safety and quality. The instrument’s scores can serve as indicators to track an organization’s progress toward PCC adoption. For instance, consistently poor ratings in areas, such as flexible care or information sharing, may lead to focused development initiatives to reduce communication breakdowns, which are a major contributor to unfavorable patient outcomes [34]. By integrating the PCCI into routine organizational audits, institutions can monitor long-term gains and evaluate how PCC-centered initiatives affect patient satisfaction, safety outcomes, and overall care quality.

Finally, the PCCI transcends its role as a measurement tool; it transforms into an effective catalyst for change by encouraging continuous learning, strengthening therapeutic alliances, and integrating PCC principles into practice and education. Its systematic use can immediately enhance clinical safety, foster a culture of empathy and collaboration among healthcare professionals, and improve patient experiences.

#### 4.3.2. Implications for Practice and Education: Insights Based on Each PCCI Concept

The eight distinct concepts of the PCCI provide a comprehensive framework for understanding and enhancing PCC. Each concept offers unique insights into provider perceptions and presents specific pathways for educational development and improved practice.

##### Concept 1: Respect and Empathy

Analysis of item-level scores reveals interesting nuances; for example, high scores on the item “Interacting with patients with respect as individuals,” coupled with lower scores on the item “Understanding the intentions of patients and their families through not only their language but also their facial expressions, attitudes, and behaviors,” might suggest that providers grasp the concept of respect but lack proficiency in nonverbal communication or deeper contextual understanding. Similarly, low scores on the item “Helping patients find meaning from their illness experiences” could indicate a need for education in narrative medicine, existential support, and moving beyond purely biomedical approaches.

Such findings align with a previous study [50] on the Consultation and Relational Empathy (CARE) Measure to assess empathy as a relational and dynamic process within clinical consultations. The CARE Measure emphasizes both verbal and nonverbal aspects of empathic communication, underscoring the importance of understanding patients’ experiences holistically. Discrepancies between respecting opinions and recognizing patient capabilities may highlight a gap in empowering patient agency. Higher scores for this concept are expected to have a profound impact [50].

Moreover, Babaii et al. [51] elaborate on this by highlighting that truly empathetic nurse-patient communication extends beyond surface-level exchanges and involves a profound emotional presence, active listening, and an intentional effort to grasp the patient’s inner world. Their study underscores that empathy involves not just recognition of emotional states but also a deep, contextualized understanding cultivated through attentiveness to both verbal expressions and subtle nonverbal cues. When healthcare professionals consistently demonstrate respectful and empathetic interactions, patients feel heard, recognized, and valued, leading to increased satisfaction and trust. Such positive relationships encourage open dialog and greater patient disclosure [51]. A pervasive caring attitude among professionals can cultivate a patient-centered organizational culture [52], potentially reducing complaints and disputes and freeing up valuable provider time previously spent on conflict resolution.

##### Concept 2: Partnership and Trust

High scores reflect the significant efforts of providers who actively promote patient and family involvement, viewing them as integral members of the care team, thus laying a strong foundation for shared decision-making. Conversely, low scores may indicate a more traditional, paternalistic approach or a lack of skills in genuinely engaging patients as partners. A specific deficit in “Helping patients express their opinions” could highlight a critical need for communication training focused on elicitation techniques. A notable discrepancy between intellectually “recognizing” patients as vital members and actively “encouraging” their participation can signal a gap between belief and actionable behavior. Improving scores for this concept is anticipated to empower patients, allowing them to actively participate in their treatment and health management. This would foster improved self-management behaviors, such as medication adherence and fall prevention, ultimately enhancing quality of life [53]. Elevated trust between the healthcare team and the patient can boost provider awareness and motivation regarding their role in patient empowerment [54]. Furthermore, enhanced collaboration among healthcare teams can lead to smoother operations, reducing medical errors and improving overall communication [55].

##### Concept 3: Individualization and Consideration for Diversity

High scores signify providers’ deep appreciation for individualized care and active pursuit of related efforts, demonstrating cultural competence and respecting personal values. Low scores may indicate a “one-size-fits-all” approach, possibly stemming from time constraints, lack of awareness, or insufficient training in cultural competence. Specific low scores on items related to “personality and characteristics” or “lifestyles and values” suggest a need to move beyond purely clinical considerations in training. Consistency in scores across items, such as equally strong consideration for “preferences” and “being sensitive to concerns,” can pinpoint areas of nuanced training needs [56]. Tailoring care to the individual is expected to significantly improve patient satisfaction, as indicated by higher scores for this concept. Respecting patients’ cultural and religious values intrinsically promotes trust and deepens collaboration [57,58]. This approach encourages healthcare professionals to become more open-minded and adaptable, thereby elevating the overall quality of care delivered [59].

##### Concept 4: Shared Decision-Making

High scores indicate that providers actively facilitate patient choice and provide comprehensive, understandable information. Conversely, low scores might indicate a paternalistic stance, a lack of confidence in guiding complex patient decisions, or insufficient skills in simplifying intricate medical information. A particular deficiency in “Providing patients with information about complications and treatment effectiveness” might highlight a direct need for enhanced communication training [60]. A disconnect between merely “providing information” and genuinely “empowering choice” suggests a critical gap in the decision-making process [61]. By enhancing shared decision-making support, patients receive necessary information in an easily assimilable format, enabling them to decide what actions they can take to improve their health and prevent disease [62]. Thus, patients can receive their desired medical care, in turn increasing their satisfaction [63,64,65]. Furthermore, patients assume a greater role in managing their own health, fostering a heightened sense of self-management [66] and self-confidence. This collaborative approach transforms treatment from being perceived as imposed to being a shared endeavor, cultivating an equitable relationship between patients and providers. It also strengthens informed consent processes [67,68], potentially reducing the risk of medical malpractice and litigation [69].

##### Concept 5: Emotional and Psychological Support

High scores on items related to this concept indicate that healthcare providers prioritize and actively facilitate patients’ emotional and psychological well-being [70], effectively leveraging social support networks. Conversely, low scores may suggest that emotional and psychological support is deprioritized compared with physical care, or that providers feel ill-equipped to offer the same. Specifically, a low score on the item “Allowing significant others to participate in providing emotional support to the patient” could underscore a need to educate providers on the profound value of family presence and involvement. Furthermore, while the item emphasizing “what they can do” is linked to empowerment, scores might also shed light on whether the “being present” and comforting aspect of emotional support is less recognized or emphasized in practice. Thus, robust emotional and psychological support allows patients suffering from illness or disability to accept their current situation and enhance their resilience to move forward, helping them navigate their health journey with greater security [19,26] and confidence [70].

Further, PCC—emphasizing the values, preferences, and holistic well-being of an individual—has become a transformative approach in the rapidly changing healthcare context. Emotional and psychological care, which goes beyond clinical interventions to recognize and foster the patient’s inner world, their identity, fears, goals, and sense of worth, is at the heart of PCC.

Healthcare professionals who use PCC cultivate therapeutic connections by acknowledging the feelings of promoting meaningful communication with and actively listening to patients. These interpersonal components foster a nurturing atmosphere in which patients experience psychological empowerment and emotional security. Barry and Edgman-Levitan [71] assert that open communication and emotional support are necessary for shared decision-making, which is a fundamental component of PCC. Patients are more satisfied and engaged when they can voice concerns, define values, and actively participate in their care. Thus, emotional support is recognized as an important element in safe, effective patient and family care [72]. When clinical and emotional components are included in therapy, the patient experience is improved [73,74].

##### Concept 6: Comprehensive Care and Holistic Perspective

High scores indicate that providers demonstrate a strong understanding and application of this model, seamlessly integrating various aspects of a patient’s life into their care plan [75]. By contrast, low scores could indicate a predominant focus on the biomedical model, in which care is primarily driven by physical symptoms and diagnoses, potentially neglecting broader life impacts. Low scores on items related to “spiritual aspects” or “leveraging patients’ strengths” might indicate specific educational deficits among providers. A discrepancy between merely “understanding” the holistic concept and actually “supporting” patients in living meaningful lives or leveraging their strengths signals a gap in practical application. Therefore, providing comprehensive care from a holistic perspective is anticipated to lead to care that considers the patient’s entire life context [76], promoting a smoother transition to life after discharge. This approach helps patients experience a life true to their values, finding deeper meaning and purpose [77]. For healthcare professionals, recognizing the importance of care that extends beyond merely curing illness can make their work more rewarding and impactful [78].

PCC—a practice philosophy that values each person’s individuality and their right to actively participate in their treatment—has become more well-known in the modern healthcare environment. The holistic approach and comprehensive care concepts are fundamental to PCC because they direct medical practitioners to treat the full range of a patient’s needs rather than just managing their illness.

Comprehensive care refers to a coordinated and ongoing approach that attends to a patient’s physical, psychological, social, and preventive health requirements at different phases of life and illness. It lessens fragmentation by supporting transitions across care venues and integrating acute, chronic, and preventive care services [21]. Offering more services is only one aspect of comprehensive care within person-centered frameworks; another is providing integrated care that is in line with the individual’s objectives, values, and life experiences.

The holistic perspective is based on a philosophy that guides the provision of care within an all-encompassing approach. It sees the individual as a complete being, with interrelated physical, emotional, mental, social, cultural, and spiritual elements influencing their health. According to prior research [22], a holistic approach is essential to person-centered treatment because it recognizes that people’s well-being goes beyond their biological functioning and respects their complete humanity.

##### Concept 7: Effective Information Sharing with Care Recipients

High scores reflect the competence of providers who excel at promoting health literacy and delivering information clearly, accessibly, and tailored to individual patient needs and learning styles. Low scores may reveal challenges in simplifying complex medical jargon, effectively assessing patient comprehension, or competently teaching self-management skills. Specifically, a low score on the item “Educating patients on how to approach their health challenges in a way that is easy for them to implement” directly points to a need for more practical education strategies. A notable difference between passively “providing” information (e.g., handing out leaflets) and actively “educating” or engaging in a two-way “talking together” dialog suggests a need for more interactive communication training. Effective information sharing facilitates appropriate responses to patient questions, improves health literacy [79], and fosters better self-management capabilities [80]. Crucially, it reduces ambiguous explanations and misunderstandings that can lead to distrust and conflict, thereby preventing potential problems between medical professionals and patients [81].

PCC is made possible by the essential practice of sharing effective information with care recipients. Fundamentally, PCC entails acknowledging people as active participants in their health journey, which is made feasible by open, considerate, and meaningful communication between patients and healthcare providers. Effective information sharing empowers patients to understand their diseases, make informed decisions, and participate in care decisions that are consistent with their beliefs and preferences [82].

Effectively sharing information involves more than just delivering facts. It entails adjusting the timing, topic, and style of communication to the patient’s emotional state, reading level, and cultural background [34]. For care receivers to feel comfortable asking questions, voicing concerns, and clearing up any doubts, health providers must establish a dialogic environment. People are thus empowered, and their autonomy is affirmed, which are important aspects of PCC [71].

##### Concept 8: Flexible Care

High scores indicate providers’ skills in strong, adaptive, and responsive care delivery, regularly re-evaluating and adjusting care plans based on patients’ responses and changing circumstances [83]. This suggests a dynamic rather than static approach to care. Conversely, low scores could indicate a rigid, protocol-driven approach that fails to adequately adapt to individual patient needs or a lack of systematic processes for ongoing assessment and review. In particular, a low score for the item examining a provider’s task of “continually reviewing the patient’s goals” might highlight a need for improved care planning and review processes within the healthcare system. Flexible and responsive care, as measured by this concept, is perceived as care that deeply incorporates the patient’s evolving thoughts and feelings [84], leading to heightened patient satisfaction [85]. Insights gained through continuous reflection on this adaptive process can be directly utilized in future care, leading to ongoing quality improvement. The ability to respond adaptively assures patients that their care is uniquely tailored, thereby increasing trust and satisfaction. By moving away from a one-size-fits-all approach, flexible care grants healthcare providers greater discretion and adaptability, ultimately leading to a more resilient and patient-responsive care system.

The fundamental tenet of PCC is that patients are active participants with unique goals and a range of life experiences, rather than passive recipients of care. This viewpoint necessitates adaptability in both the organization and the delivery of care. According to McCormack and McCance [22], a person-centered approach necessitates being sensitive to the lived experience of the individual and having the flexibility to modify care plans in real time in response to the patient’s top priorities.

Flexible care enables medical practitioners to adjust their methods to the preferences of the patient by changing clinical judgments, communication tactics, care routines, and scheduling. For instance, a patient suffering from a chronic illness may need care tailored to their daily energy levels or family obligations. According to Santana et al. [21], flexible care allows for prompt modifications that preserve a person’s autonomy and dignity, whereas a protocol-driven approach might not be able to handle such variability.

While this conceptual framework and the PCCI provide a vital tool for understanding and improving PCC, it is important to acknowledge the present study’s limitations and outline a clear path for future research to fully establish the instrument’s psychometric properties and utility.

### 4.4. Limitations

Despite the rigorous content validation process, this study has several limitations that warrant consideration.

First, the current stage of validation is limited to face and content validity. While these steps are essential, they do not fully establish the psychometric properties of the PCCI. Further statistical analyses, including exploratory and confirmatory factor analyses, are needed to confirm the proposed eight-concept structure and ensure construct validity. Items that met the I-CVI threshold of 0.70 or higher were retained for a subsequent large-scale survey. In future studies, these items will be further refined or potentially eliminated based on their psychometric properties, including item-total correlations and factor loadings, to create a more robust final scale.

Second, the expert panel, while diverse in terms of profession, was predominantly composed of nurses. This limited representation of other disciplines, such as physical therapists (n = 2) and physicians (n = 1), may introduce a professional bias and impact the generalizability of the content validation. We recommend that future validation studies include a more balanced representation across medicine, social work, and other healthcare fields to minimize this bias and broaden the instrument’s applicability.

Third, as a self-report instrument, the PCCI may be subject to social desirability bias, where participants tend to report what they perceive as desirable rather than their actual practices or beliefs regarding PCC.

Finally, the present study did not include pilot testing with the target population beyond the expert review. Pilot testing could provide valuable insights into item clarity and how respondents interpret the instrument in a real-world setting.

### 4.5. Future Studies

Building upon the foundational work of this study, several avenues for future research are recommended to fully establish and leverage the PCCI.

Psychometric Validation: The immediate next step is to conduct a large-scale psychometric evaluation of the PCCI using a diverse sample of healthcare providers. This could involve statistical analyses, such as exploratory and confirmatory factor analyses, to definitively establish the instrument’s construct validity and internal consistency (reliability) across all eight concepts.

Criterion and Predictive Validity: Future research should investigate the criterion validity of the PCCI by correlating its scores with other established measures of PCC or related constructs (e.g., patient satisfaction scores, communication effectiveness, and empathy scales). Furthermore, examining the tool’s predictive validity, how well scores predict tangible patient outcomes or quality of care indicators, would significantly strengthen its utility.

Cross-Cultural Adaptation and Validation: Studies should adapt and validate the instrument in various cultural and healthcare settings to confirm its applicability beyond the initial development context. This would involve rigorous translation processes and re-validation of psychometric properties in new populations.

Longitudinal Studies and Intervention Evaluation: The PCCI can serve as a valuable tool for longitudinal studies to track changes in providers’ PCC perceptions over time, particularly in response to educational interventions or policy changes. Future research could utilize the PCCI to evaluate the effectiveness of PCC training programs by measuring changes in provider scores before and after interventions.

Qualitative Exploration: Complementing quantitative data with qualitative methods (e.g., interviews and focus groups with providers and patients) can provide richer insights into the nuances of each PCC concept and the lived experiences of receiving PCC. This can help refine existing items or identify new dimensions for future iterations of the PCCI.

Linking Provider Perceptions to Patient Experiences: Ultimately, research should explore the direct relationship between provider scores on the PCCI and actual patient experiences and outcomes. This could involve multi-source data collection, comparing provider self-assessments with patient reports of PCC.

## 5. Conclusions

The developed PCCI is a 37-item tool that evaluates healthcare providers’ competence in delivering PCC across eight key domains. Validated through a rigorous Delphi process, the PCCI offers a structured and valid framework for assessing essential PCC competencies.

The PCCI contributes to the literature by addressing the limitations of existing profession-specific tools, providing a cross-disciplinary resource that reflects the holistic and relational nature of PCC. From a patient’s perspective, the PCCI highlights competencies such as respect, empathy, and shared decision-making, which are directly linked to satisfaction and trust. For healthcare providers, the tool reinforces relational and communication skills crucial for therapeutic partnerships and holistic practice.

By aligning professional competencies with patient priorities, the PCCI has the potential to improve quality of care, enhance patient outcomes, and foster a culture of empathy and collaboration in healthcare.

## Figures and Tables

**Figure 1 nursrep-15-00355-f001:**
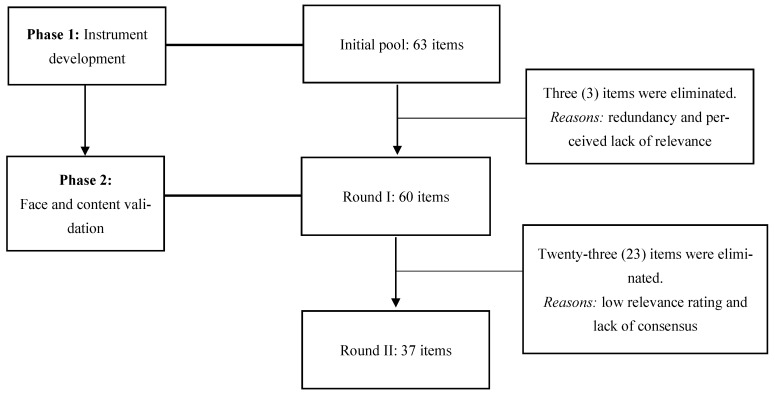
Item development process.

**Table 1 nursrep-15-00355-t001:** Expert panel characteristics.

Characteristics	Respondent (n = 10)
	Frequency (%)
**Gender**	
Male	5 (50)
Female	5 (50)
**Age, years**	
30–39	2 (20)
40–49	3 (30)
50–59	3 (30)
60–69	2 (20)
**Working period in healthcare**	
10–29 years	5 (50)
More than 30 years	5 (50)
**Discipline**	
Medicine	1 (10)
Nursing	7 (70)
Physical Therapy	2 (20)
**Institutional affiliation/Occupation**	
University hospital staff	1 (10)
University staff	7 (70)
JICA consultant	1 (10)
Director of nursing	1 (10)
**Educational attainment**	
Master’s	1 (10)
PhD	9 (90)

JICA: Japan International Cooperation Agency.

**Table 2 nursrep-15-00355-t002:** Items of the developed PCCI.

Concept	No.	Questions	Median	Min.	Max.	I-CVI	UA
C1	1	Interacting with patients with respect as individuals	8	7	9	1	1
C4	39	Sharing patient decision-making with the care team	8	7	9	1	1
C6	50	Understanding the person holistically, considering not only their illness but also related physical, psychological, social, cultural, and spiritual aspects	8	7	9	1	1
C1	2	Communicating with patients to understand their hopes and dreams	7.5	5	9	0.9	0
C5	10	Attempting to understand and communicate what the patient is feeling and communicating	8	5	9	0.9	0
C6	17	Support focused on improving quality of life by leveraging patients’ strengths	7	5	8	0.9	0
C3	28	Providing support based on the treatment needs of patients	8	5	9	0.9	0
C4	37	Empowering patients to make informed decisions	8	5	9	0.9	0
C4	40	Respecting the patient’s choices and decisions regarding their life	7	5	9	0.9	0
C4	41	Providing patients with information about complications and treatment effectiveness	7.5	5	9	0.9	0
C6	51	Supporting patients in living the lives that are right for them	7.5	5	9	0.9	0
C8	63	Continually reviewing the patient’s goals and the plan for achieving those goals	8	5	9	0.9	0
C5	5	Listening carefully to the patient’s experience	7.5	5	9	0.8	0
C1	9	Understanding the intentions of patients and their families through not only their language but also their facial expressions, attitudes, and behaviors	8	5	8	0.8	0
C2	13	Helping patients express their opinions in their care	7.5	3	9	0.8	0
C4	36	Empowering patients to choose treatments that fit their lifestyles and values	7.5	4	9	0.8	0
C5	19	Allowing significant others to participate in providing emotional support to the patient, if necessary	7	5	8	0.8	0
C7	56	Providing information in a way that patients can understand	7.5	3	9	0.8	0
C7	57	Educating patients on how to approach their health challenges in a way that is easy for them to implement	7	5	9	0.8	0
C8	59	Providing prompt and appropriate support according to the patient’s changes and situation	7	5	8	0.8	0
C1	3	Empathizing with the patient’s feelings and providing compassionate care	8	5	8	0.7	0
C8	6	Respecting the patient’s opinion	7	3	8	0.7	0
C2	8	Recognizing patients as persons who have the resources and capabilities to solve their problems	7	3	8	0.7	0
C2	11	Recognizing the patient as a vital member of the care team	6.5	5	8	0.7	0
C2	12	Encouraging patients to participate in their care	6	1	8	0.7	0
C2	15	Deepening the common understanding of the patient’s condition (problem) with the patient and their family	7	5	8	0.7	0
C3	24	Considering the patient’s most important wishes and health concerns	7	1	8	0.7	0
C3	29	Providing care that considers the patient’s values and beliefs	8	3	9	0.7	0
C3	31	Providing support tailored to the patient’s personality and characteristics	7.5	3	9	0.7	0
C8	33	Helping patients find meaning from their illness experiences	6.5	3	8	0.7	0
C4	34	Empowering patients to make decisions for themselves	7.5	4	9	0.7	0
C5	45	Providing comprehensive care to meet the physical, psychological, and social needs of patients	8	2	8	0.7	0
C6	49	Supporting patients with realistic health and life goals	7	3	8	0.7	0
C7	52	Providing patients with information and methods for self-care	7	3	8	0.7	0
C7	53	Talking together with patients about what they can do to improve their health and prevent illness	7	5	8	0.7	0
C7	54	Providing patient-specific education and care	7	3	9	0.7	0
C7	55	Providing patients with adequate and appropriate information	7	3	9	0.7	0
C7	20	Explaining the health condition to the patient and their family in an easy-to-understand manner	6.5	4	8	0.6	0
C3	22	Providing care taking into consideration the patient’s preferences, lifestyle, values, etc.	6.5	3	8	0.6	0
C3	25	Being sensitive to the patient’s concerns, wishes, and priorities	7	2	8	0.6	0
C3	27	Providing care adapted to the patient’s priorities	7	1	8	0.6	0
C3	32	Providing individualized, goal-oriented assistance	6	1	8	0.6	0
C4	38	Providing information to help patients decide what healthcare services they need	8	3	9	0.6	0
C5	16	Helping patients understand what they can do and making them feel more confident	7	5	9	0.6	0
C6	47	Providing care to help patients live comfortably and recover physically	6	3	8	0.6	0
C1	7	Making time to intentionally listen to patients talk	5.5	4	9	0.5	0
C5	18	Providing support to patients with care and consideration for their feelings and situations	6	3	8	0.5	0
C3	23	Understanding the patient’s values and habits	6	3	8	0.5	0
C3	30	Providing care tailored to individual needs	6	3	9	0.5	0
C5	43	Understanding the patient’s emotional needs	5.5	3	8	0.5	0
C5	44	Responding to the patient’s spiritual needs	5.5	3	8	0.5	0
C8	58	Modifying care to suit the patient’s situation, if necessary	6	3	8	0.5	0
C8	60	Responsive and flexible to patient needs	5.5	2	8	0.5	0
C8	62	Paying attention to the patient’s reactions to adapt to their situation at any given time	4.5	3	8	0.4	0
C6	46	Considering the patient’s physical needs	5	3	9	0.3	0
C6	48	Understanding the social needs of patients	5	2	8	0.3	0
C8	61	Adapting to the situation, paying attention to the patient’s reactions (physical reactions, language, and emotional cues)	5	3	8	0.3	0
C7	21	Keeping the patient and their family informed about changes in health status	4.5	3	7	0.2	0
C3	26	Prioritizing the patient’s daily living preferences	4.5	3	6	0.2	0
C4	14	Sharing patient decision-making among the care team	5	5	8	0.1	0

C: Concept; I-CVI: Item-level Content Validity Index; UA: Universal Agreement. The table presents details of the concepts, retained items, I-CVIs, and median ratings. Three items achieved universal agreement among the experts (I-CVI = 1.0). The final 37-item PCCI had I-CVI ≥ 0.7. Among the retained items, 12 achieved very high relevance, with an I-CVI exceeding 0.90.

## Data Availability

The data presented in this study are available upon request from the corresponding author. These data are not publicly available due to privacy restrictions.

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
