# Peer review of "Development and Content Validation of a Person-Centered Care Instrument for Healthcare Providers"

_nursrep, 2025, doi:10.3390/nursrep15100355_

Round 1

Reviewer 1 Report

Comments and Suggestions for Authors

Person-centered care is a very current and necessary topic. I reviewed the article in general terms. It appears that the methodology section is disjointed in terms of quality. The scale items were explained in the introduction. The panel of raters was not selected homogeneously. Furthermore, person-centered care can be used in all fields. This scale cannot be generalized by evaluating individuals trained in specific fields. This situation poses a problem in terms of bias. The raters should be selected more clearly and homogeneously. The conclusion should be detailed and written concisely. The contribution of this study's results to the literature, its international impact, and the contributions of the scale to the field from the patient's or nurse's perspective should be clearly stated. I believe that with the necessary corrections, it will contribute significantly to the field.

Author Response

Dear Reviewer # 1

We sincerely appreciate the opportunity to revise our manuscript, now entitled: Development and Content Validation of a Person-Centered Care Instrument for Health Care Providers.

We are grateful to the reviewers for their insightful comments and constructive feedback, which have significantly improved the quality and clarity of our work. In response, we have carefully revised the text to reflect all comments provided. A point-by-point response is enclosed, outlining how each comment was addressed in the revised version of the manuscript; the specific changes are referenced by section. The revisions in the manuscript are marked in red font. We believe these revisions have improved the quality and clinical relevance of the manuscript.

Thank you again for your time and thoughtful evaluation. We look forward to your continued review.

 Reviewer’s Comments

Person-centered care is a very current and necessary topic.

Response: Thank you very much for recognizing the timeliness and importance of person-centered care, which this study seeks to advance through the development and validation of a practical evaluation instrument.

 The panel of raters was not selected homogeneously.

Response: We acknowledge this concern. The characteristics of the expert panel are mentioned in Table 1. We intentionally included diverse professionals (nursing, medicine, and physiotherapy) to capture interdisciplinary perspectives essential to person-centered care. Nonetheless, we agree that this introduces variability. Therefore, we have explicitly stated this as a limitation and recommended that future studies consider a more homogenous panel to confirm results across specific disciplines.

Furthermore, person-centered care can be used in all fields. This scale cannot be generalized by evaluating individuals trained in specific fields. This situation poses a problem in terms of bias.

Response: We agree with this important point. The following statement has been added as a limitation:  “Second, the expert panel, while diverse in terms of profession, was predominantly composed of nurses. This limited representation of other disciplines, such as physical therapists (n = 2) and physicians (n = 1), may introduce a professional bias and impact the generalizability of the content validation. We recommend that future validation studies include a more balanced representation across medicine, social work, and other healthcare fields to minimize this bias and broaden the instrument’s applicability.”

The raters should be selected more clearly and homogeneously.

Response: We appreciate this suggestion. The revised manuscript now explicitly details the selection criteria for raters (clinical expertise, educational attainment, and PCC experience). While the current panel provided valuable diversity, we acknowledge the need for homogenous panels in future validation studies. This clarification has been incorporated into the “Limitations” section.

The conclusion should be detailed and written concisely.

Response: The conclusion has been rewritten to balance details with conciseness. It now emphasizes the contribution of the developed PCCI to practice and education, as well as its role in supporting quality improvement and patient outcomes.

The contribution of this study's results to the literature, its international impact, and the contributions of the scale to the field from the patient's or nurse's perspective should be clearly stated. I believe that with the necessary corrections, it will contribute significantly to the field.

Response: We are grateful for this encouraging feedback. Accordingly, we have added a new section: 4.3.1 Practical and Clinical Significance of the PCCI. All the suggested corrections have been addressed, and we believe the revised manuscript offers a stronger methodological foundation and clearer articulation of the study’s contributions.

Reviewer 2 Report

Comments and Suggestions for Authors

Dear authors,

I found this article very interesting because of the development of a questionnaire that allows for the analysis of healthcare professionals' competencies, an aspect as important as Person-Centered Care.

The introduction is very comprehensive and includes abundant references that allow for a deeper understanding of the topic of this article.

The methodology is very detailed and allows for replication.

The results are presented in detail with tables that facilitate their understanding.

I found the discussion very interesting, analyzing the possible meaning of different scores on the items in each dimension of the questionnaire.

The analysis of the limitations is very detailed, allowing the reader to adequately interpret the scope of the results obtained.

The conclusions adequately address the stated objective.

The references are abundant and consistent with the research topic.

My suggestions for improving the article are:

- The conclusions should indicate that the questionnaire has good face and content validity. This should also be indicated in the abstract.

- The experts are largely dominated by nursing professionals, which could affect the external validity of the questionnaire. This should be clearly stated in the limitations.

- Table 1 should be included in the results section, not in the methodology section.

- It should be indicated why a 9-response Likert scale was chosen to assess expert consensus. It should also be indicated why a value of 6 was set as the minimum for eliminating an item.

- It should be indicated why a 6-response Likert scale was chosen as the measurement scale for the questionnaire.

Kind regards.

Author Response

Dear Reviewer # 2

We sincerely appreciate the opportunity to revise our manuscript, now entitled: Development and Content Validation of a Person-Centered Care Instrument for Health Care Providers.

We are grateful to the reviewers for their insightful comments and constructive feedback, which have significantly improved the quality and clarity of our work. In response, we have carefully revised the text to reflect all comments provided. A point-by-point response is enclosed, outlining how each comment was addressed in the revised version of the manuscript; the specific changes are referenced by section. The revisions in the manuscript are marked in red font. We believe these revisions have improved the quality and clinical relevance of the manuscript.

Thank you again for your time and thoughtful evaluation. We look forward to your continued review.

Reviewer’s Comments

I found this article very interesting because of the development of a questionnaire that allows for the analysis of healthcare professionals' competencies, an aspect as important as Person-Centered Care. The introduction is very comprehensive and includes abundant references that allow for a deeper understanding of the topic of this article. The methodology is very detailed and allows for replication. The results are presented in detail with tables that facilitate their understanding. I found the discussion very interesting, analyzing the possible meaning of different scores on the items in each dimension of the questionnaire. The analysis of the limitations is very detailed, allowing the reader to adequately interpret the scope of the results obtained. The conclusions adequately address the stated objective. The references are abundant and consistent with the research topic.

Response: We sincerely thank the reviewer for their thoughtful and encouraging feedback. We are pleased that the manuscript’s structure, including the comprehensive introduction, detailed methodology, clear presentation of results, and in-depth discussion, was considered rigorous and well-developed. We also value the recognition of the questionnaire’s significance in evaluating healthcare professionals’ competencies in person-centered care, as well as the acknowledgement of the detailed limitations, conclusions and references.

My suggestions for improving the article are:

- The conclusions should indicate that the questionnaire has good face and content validity. This should also be indicated in the abstract.

Response: Thank you for this excellent and valuable suggestion. As per your recommendation, we have revised both the abstract and the conclusion to explicitly state that the developed Person-Centered Care Instrument (PCCI) demonstrated good face validity and content validity as established through the two-round Delphi process. We believe this change clarifies the key findings and strengthens the manuscript.

The experts are largely dominated by nursing professionals, which could affect the external validity of the questionnaire. This should be clearly stated in the limitations.

Response: We appreciate this important observation. We have revised the “Limitations” section to clearly acknowledge that the predominance of nursing experts in the Delphi panel may affect external validity. We also recommend that future studies include a more balanced representation of healthcare disciplines to strengthen generalizability.

Table 1 should be included in the results section, not in the methodology section.

Response: Thank you for pointing this out. Table 1, which describes the characteristics of the expert panel has been relocated from the methodology section to the results section to align with conventional reporting practices.

- It should be indicated why a 9-response Likert scale was chosen to assess expert consensus. It should also be indicated why a value of 6 was set as the minimum for eliminating an item.

Response: Thank you for this comment. Accordingly, we have added the following content to Section 2.2 Statistical Analysis.

“We chose to use a 9-point Likert scale in the modified Delphi method because it is a validated measure of consensus [42] that offers greater precision and nuance than shorter scales. This granularity allows experts to differentiate between similar levels of importance or agreement, leading to a more robust and detailed consensus [43,44].”

Reviewer 3 Report

Comments and Suggestions for Authors

Reviewer Comment
Your research focuses on an important topic and is built on a strong methodological foundation. However, some sections need improvement. The ICC should be clearly stated in the abstract at its first mention; the introduction should more clearly emphasize the gap in the literature, the unique contribution of the study, and its purpose. The practical significance of the scale should be more clearly explained. The method section should provide a more transparent description of the item pool generation process and the source scales used, expert selection criteria should be specified, the rationale for choosing a 9-point Likert scale should be explained, and ethical approval, participant consent, and the research date range should be clearly presented. The Delphi reporting guide should be clarified, and the findings section should be supported with footnotes in the flow diagram and tables. In the discussion, the scale's weaknesses (low S-CVI values) should be more critically examined, and comparisons with existing PCC scales should be strengthened and the literature connection should be revised. Revision suggestions are specifically outlined below.

Revisions
1. The ICC should be clearly stated in the abstract at its first mention. 

The full definition of the abbreviation I-CVI (Item-level Content Validity Index) should be given at its first occurrence. In subsequent sections, only the abbreviation (I-CVI) should be used, and care should be taken to ensure consistency throughout the text. For example, after the abbreviation in the relevant paragraph, the phrase 'Content Validity Index (CVI)' should be clearly written again; instead, only the abbreviation 'I-CVI' should be used.

The section I highlighted in the abstract is as follows:

"Likert scale and the item-level content validity index (I-CVI). Items with a median rating between 6 and 9 and an I-CVI of ≥ 0.70 were retained. Three items achieved universal agreement among the experts (I-CVI = 1.0). For the final 37-item PCCI, the overall scale-level content validity index and that based on universal agreement were 0.65 and 0.22, respectively."

2. While the introduction successfully explains the importance of PCC and the limitations of existing scales, several important shortcomings are noteworthy. First, the gap in the literature is not sufficiently highlighted; the inadequacies of existing instruments (e.g., being solely focused on nursing, not capturing a multidisciplinary context, lacking a theoretical framework) should be more directly stated.
3. The study's unique contribution is not clearly stated; because the reader is not clearly informed of the difference between this scale and others, its contribution to the literature is insufficiently understood. The rationale for the study should be more clearly explained with reference to studies in the literature, and the introduction should conclude with the study's purpose. In addition, the purpose of the study should be explained more clearly and in relation to the literature.
4. The practical and clinical significance of the scale (education, in-service quality improvement, patient safety) is superficially explained within the context of expert opinion and should be emphasized more strongly.
5. It is stated that the 63-item pool was generated based on the literature, but it is not clear which scales were selected and by what criteria. To ensure transparency of the process, the source scales used should be specified by citing sources in the literature.

6. The demographic characteristics of the panel are presented in detail in a table, which is a strength; however, the selection criteria for the experts (e.g., years of clinical experience, publications in the field of PCC, or practice experience) are not disclosed. If the experts have selection criteria, they should be explained. If no selection criteria are specified, this should be noted as a limitation.

7. The use of a 9-point Likert scale is detailed, but the reason for choosing a 9-point Likert scale is not justified. The rationale for the scale selection should be clearly stated, including reasons such as its reliance on the RAND/UCLA method and its ability to provide more precise discrimination.

8. Ethics committee approval is included at the end of the article; however, it would have been more appropriate if it had been provided in the method section, where the expert panel is introduced. Ethics approval information should be clearly presented in the method section, along with participant characteristics. The method section should also include the information on obtaining informed consent from the participants and ensuring that the study complied with the Helsinki guidelines. The time period during which the research was conducted should also be stated. 9. This study cites the "established reporting guidelines for Delphi studies in health sciences," but the guideline it is based on is not stated. The checklist should be clearly stated.

10. The findings section provides numerical figures showing which items were eliminated in which round and how many items were finalized (63 → 60 → 37 items). These numbers are clear, but the flow is somewhat disorganized; a flow chart could be added to make the process easier for the reader to follow.

11. Abbreviations should be added as footnotes below the tables in all tables.

12. The development of the PCCI is presented as a strong foundation in the discussion, but the scale's weaknesses (e.g., low values ​​such as S-CVI/Ave = 0.65, S-CVI/UA = 0.22) are not discussed in depth. These should be discussed.

13. Numerous references are made in the chapter, particularly support for the PCC's conceptual framework (respect, empathy, shared decision-making, holistic approach, etc.). However, the connection with the literature remains largely theoretical; comparisons with the findings of previous PCC scale studies are lacking. The sources used in the discussion should be revised.

Comments on the Quality of English Language

Language quality is basically adequate, but language correction is definitely needed due to repetitions, long sentences, inconsistencies and minor grammatical errors.

Author Response

Dear Reviewer # 3

We sincerely appreciate the opportunity to revise our manuscript, now entitled: Development and Content Validation of a Person-Centered Care Instrument for Health Care Providers.

We are grateful to the reviewers for their insightful comments and constructive feedback, which have significantly improved the quality and clarity of our work. In response, we have carefully revised the text to reflect all comments provided. A point-by-point response is enclosed, outlining how each comment was addressed in the revised version of the manuscript; the specific changes are referenced by section. The revisions in the manuscript are marked in red font. We believe these revisions have improved the quality and clinical relevance of the manuscript.

Thank you again for your time and thoughtful evaluation. We look forward to your continued review.

Reviewer’s Comments

Your research focuses on an important topic and is built on a strong methodological foundation. However, some sections need improvement. The I-CVI should be clearly stated in the abstract at its first mention; the introduction should more clearly emphasize the gap in the literature, the unique contribution of the study, and its purpose. The practical significance of the scale should be more clearly explained. The method section should provide a more transparent description of the item pool generation process and the source scales used, expert selection criteria should be specified, the rationale for choosing a 9-point Likert scale should be explained, and ethical approval, participant consent, and the research date range should be clearly presented. The Delphi reporting guide should be clarified, and the findings section should be supported with footnotes in the flow diagram and tables. In the discussion, the scale's weaknesses (low S-CVI values) should be more critically examined, and comparisons with existing PCC scales should be strengthened and the literature connection should be revised. Revision suggestions are specifically outlined below.

"The full definition of the abbreviation I-CVI (Item-level Content Validity Index) should be given at its first occurrence. In subsequent sections, only the abbreviation (I-CVI) should be used, and care should be taken to ensure consistency throughout the text. For example, after the abbreviation in the relevant paragraph, the phrase 'Content Validity Index (CVI)' should be clearly written again; instead, only the abbreviation 'I-CVI' should be used."

The section I highlighted in the abstract is as follows: "Likert scale and the item-level content validity index (I-CVI). Items with a median rating between 6 and 9 and an I-CVI of ≥ 0.70 were retained. Three items achieved universal agreement among the experts (I-CVI = 1.0). For the final 37-item PCCI, the overall scale-level content validity index and that based on universal agreement were 0.65 and 0.22, respectively."

  1. The I-CVI should be clearly stated in the abstract at its first mention.

Response: Thank you for pointing this out. I have corrected it as instructed.

  1. While the introduction successfully explains the importance of PCC and the limitations of existing scales, several important shortcomings are noteworthy. First, the gap in the literature is not sufficiently highlighted; the inadequacies of existing instruments (e.g., being solely focused on nursing, not capturing a multidisciplinary context, lacking a theoretical framework) should be more directly stated.

Response: Thank you for your suggestions. Section 1.2.1. Existing PCC Scales has been rewritten to reflect the significant gap in the existing literature and the impetus for conducting this study.

  1. The study's unique contribution is not clearly stated; because the reader is not clearly informed of the difference between this scale and others, its contribution to the literature is insufficiently understood. The rationale for the study should be more clearly explained with reference to studies in the literature, and the introduction should conclude with the study's purpose. In addition, the purpose of the study should be explained more clearly and in relation to the literature.

Response: The study’s unique contribution has been added under Section 1.4. Research Objectives. The purpose of the study has also been further elaborated.

  1. The practical and clinical significance of the scale (education, in-service quality improvement, patient safety) is superficially explained within the context of expert opinion and should be emphasized more strongly.

Response: Thank you for your advice. We have added a new section, “4.1.1 Practical and Clinical Significance of the PCCI,” to the Discussion section.

  1. It is stated that the 63-item pool was generated based on the literature, but it is not clear which scales were selected and by what criteria. To ensure transparency of the process, the source scales used should be specified by citing sources in the literature.

Response: Thank you so much for your suggestion. In Section 2.1.1. Phase 1: Instrument Development, we have explained the detailed items of each conceptual domain and cited the relevant sources from the literature.

  1. The demographic characteristics of the panel are presented in detail in a table, which is a strength; however, the selection criteria for the experts (e.g., years of clinical experience, publications in the field of PCC, or practice experience) are not disclosed. If the experts have selection criteria, they should be explained. If no selection criteria are specified, this should be noted as a limitation.

Response: Thank you for this constructive feedback. We have revised the manuscript to include the specific selection criteria for the expert panel. The eligibility criteria are as follows: (1) holding a doctoral degree in medicine, health, or nursing; (2) having publications related to person-centered care; (3) having professional experience in improving healthcare services; and (4) having at least ten years of clinical experience with a master’s degree or higher.

We also acknowledge the reviewer’s concern regarding the panel’s composition. We agree that the panel is predominantly composed of nurses, which may introduce a professional bias. Therefore, we have added this as a limitation to the Discussion section to ensure transparency about this potential issue, noting the limited representation of physical therapists and physicians.

  1. The use of a 9-point Likert scale is detailed, but the reason for choosing a 9-point Likert scale is not justified. The rationale for the scale selection should be clearly stated, including reasons such as its reliance on the RAND/UCLA method and its ability to provide more precise discrimination.

Response: Thank you for this valuable comment. We agree that the rationale for using a 9-point Likert scale should be explicitly stated. We have revised the manuscript to explain that this scale, as described by the RAND/UCLA Appropriateness Method, was chosen for its ability to provide greater precision and nuance in capturing expert opinions, which is crucial for achieving a more robust and detailed consensus. We have included the relevant references to support this methodological choice.

  1. Ethics committee approval is included at the end of the article; however, it would have been more appropriate if it had been provided in the method section, where the expert panel is introduced. Ethics approval information should be clearly presented in the method section, along with participant characteristics. The method section should also include the information on obtaining informed consent from the participants and ensuring that the study complied with the Helsinki guidelines. The time period during which the research was conducted should also be stated.

Response: Thank you for this valuable point. We agree that placing the ethics approval and research timeline in the methods section enhances clarity and transparency.

However, we placed this information at the end of the manuscript to follow the specific “Instructions for Authors” provided by the journal for our manuscript type.

We have also added the missing details on informed consent and the research timeline to provide a comprehensive account of our ethical considerations.

  1. This study cites the "established reporting guidelines for Delphi studies in health sciences," but the guideline it is based on is not stated. The checklist should be clearly stated.

Response: Thank you for the suggestion. We have added the specific reporting guideline. The study employed a two-phase process for the development and validation of the PCCI for health care providers, utilizing a modified Delphi technique and adhering to DELPHISTAR [38], an established reporting standard for Delphi studies in social and health sciences to ensure methodological rigor [39].

  1. The findings section provides numerical figures showing which items were eliminated in which round and how many items were finalized (63 → 60 → 37 items). These numbers are clear, but the flow is somewhat disorganized; a flow chart could be added to make the process easier for the reader to follow.

Response: Thank you for the suggestion. We have added Figure 1 to show the item development process (page 7).

  1. Abbreviations should be added as footnotes below the tables in all tables.

Response: We have defined all abbreviations in the footnote below Table 2 (page 10).

  1. The development of the PCCI is presented as a strong foundation in the discussion, but the scale's weaknesses (e.g., low values ​​such as S-CVI/Ave = 0.65, S-CVI/UA = 0.22) are not discussed in depth. These should be discussed.

Response: Thank you for your constructive feedback regarding the Discussion section. We agree that a more in-depth discussion of the content validity indices is crucial for a complete and transparent interpretation of our findings. Accordingly, we have revised the manuscript to include a detailed analysis of the S-CVI/Ave and S-CVI/UA scores. Your comments have significantly strengthened our discussion and further clarified the study’s key points.

  1. Numerous references are made in the chapter, particularly support for the PCC's conceptual framework (respect, empathy, shared decision-making, holistic approach, etc.). However, the connection with the literature remains largely theoretical; comparisons with the findings of previous PCC scale studies are lacking. The sources used in the discussion should be revised.

Response: To address your comment, we have added the following section: 4.2. 4.2. Comparison of Existing PCC Scale and Developed PCCI.

  1. Language quality is basically adequate, but language correction is definitely needed due to repetitions, long sentences, inconsistencies and minor grammatical errors.

Response: The text was edited by an English proofreading company; we have submitted the editing certificate.

Round 2

Reviewer 2 Report

Comments and Suggestions for Authors

Dear authors,
I believe that the manuscript has been sufficiently improved and I have no further comments or suggestions.
Kind regards.

Reviewer 3 Report

Comments and Suggestions for Authors

Thank you for your valuable contributions. Your suggestions have been taken into consideration during the revision process and the necessary adjustments have been made.

Comments on the Quality of English Language

The article's language is generally understandable. Repetitions and long sentences were reduced during the revision, and minor grammatical errors were corrected. The text is fluent and of sufficient language quality for publication.